

# Spectral response of Josephson junctions with low-energy quasiparticles

**Anna Keselman[1,2]★, Chaitanya Murthy[3], Bernard van Heck[1,4] and Bela Bauer[1]**

**1** Station Q, Microsoft Corporation, Santa Barbara, California 93106 USA
**2** Kavli Institute for Theoretical Physics, University of California,
Santa Barbara California 93106 USA
**3** Department of Physics, University of California, Santa Barbara, CA 93106
**4** Microsoft Quantum Lab Delft, Delft University of Technology,
2600 GA Delft, The Netherlands

★ akeselman@kitp.ucsb.edu

## Abstract

We study nanowire-based Josephson junctions shunted by a capacitor and take into account the presence of low-energy quasiparticle excitations. These are treated by extending conventional models used to describe superconducting qubits to include the coherent coupling between fermionic quasiparticles, in particular the Majorana zero modes that emerge in topological superconductors, and the plasma mode of the junction. Using accurate, unbiased matrix-product state techniques, we compute the energy spectrum and response function of the system across the topological phase transition. Furthermore, we develop a perturbative approach, valid in the harmonic limit with small charging energy, illustrating how the presence of low-energy quasiparticles affects the spectrum and response of the junction. Our results are of direct interest to on-going experimental investigations of nanowire-based superconducting qubits.



# 1 Introduction

Due to the macroscopic coherence of the superconducting state and their non-linearity as circuit elements, Josephson junctions are a workhorse of quantum state engineering [1]. They are the fundamental building block of superconducting qubits [2,3] like the transmon [4] and the fluxonium [5]. In these quantum engineering applications, the superconducting circuit embedding the Josephson junction [6] is operated at frequencies $\omega \sim 5$–$10$ GHz, which are much smaller than the superconducting gap of the electrodes, $\Delta/h \simeq 50$ GHz for aluminum. As a consequence, quasiparticles are not involved in the coherent dynamics of the circuit, although their presence influences the relaxation and dephasing of superconducting qubits [7–17].

New frontiers in superconducting devices force us to reconsider the role of quasiparticles in Josephson junction dynamics. Most notably, the presence of Majorana zero modes (MZMs)—topologically protected zero-energy quasiparticles that emerge at the ends of topological superconducting wires [18, 19]—can drastically affect the behavior of a superconducting circuit. MZMs are able to non-locally encode qubits [18] and, via non-Abelian braiding [20], allow fault-tolerant processing of quantum information. Therefore, they form a potential platform for topological quantum computation [21] which is actively being pursued [22]. A junction between two topological superconductors exhibits a $4\pi$-periodic Josephson effect [18, 23, 24], a hallmark feature of topological superconductivity which has been experimentally sought [25, 26]. Several practical schemes for topological quantum computation rely on the coupling of MZMs across a Josephson junction and on the use of microwave circuits for control and readout of topological qubits [27–30]. In conjunction with the growing interest in MZMs, different research groups have developed and studied superconducting devices with semiconductor-based Josephson junctions either in nanowires [31, 32] or 2DEGs [33], as well as in graphene-based heterostructures [34, 35]. These junctions, characterized by few conducting channels and potentially high transparency, can also be used for the development of qubits based on conventional Andreev bound states (ABSs) [36–40]. The presence of low-energy ABSs in nanowire-based junctions has been directly measured via microwave spectroscopy [41–43].

Understanding present and future experimental developments in this direction calls for adequate theoretical approaches that can fully incorporate the role of quasiparticles in the circuit dynamics. In most theoretical descriptions of Josephson-junction dynamics, quasiparticles are included as a fermionic bath [44] (see Ref. [45] for a recent exception). In the

case of Andreev qubits, detailed models which are amenable to an analytical approach are available in simple limits, such as that of a short Josephson junction with a single conducting channel [37, 46–48]. On the other hand, most of the theory literature treating the presence of MZMs in superconducting circuits [49–66] relies on simple toy-models with phenomenological terms representing Majorana couplings, bypassing a microscopic description of the topological phase.

In this paper, we carefully examine this problem using accurate numerical simulations of a microscopic model for a one-dimensional topological superconductor. While we confirm the applicability of simplified models in certain limits, we find that in other, experimentally relevant limits they are insufficient to describe the system's behavior. We focus on the topological phase transition and on the case of additional subgap Andreev bound states in the junction, which we expect to generically appear in wires with large spin-orbit coupling and external magnetic field. We find that the phase transition does not lead to strong signatures in the response of the capacitively shunted junction, while additional subgap Andreev states exhibit complex interplay with the plasma modes and significantly alter the response.

Our numerical simulations are based on matrix product states (MPS) [67–69]. Specifically, we use the density matrix renormalization group (DMRG) [70–72] and time-evolving block decimation (TEBD) [73–76] to compute the time-dependent charge correlation function of a nanowire Josephson junction shunted by a large capacitor (i.e. a transmon circuit) across the topological phase transition. In the frequency domain, the correlation function determines the observed spectra in a typical circuit QED (cQED) experiment, making our method suitable for direct comparison with experimental measurements. This approach allows us to determine the expected frequency spectra even close to the critical point—a regime which cannot be captured by existing toy models—and to easily include additional Andreev bound states.

In order to interpret the results of the MPS simulations, and extending previous studies [48], we also develop a simple perturbative approach which is valid in the harmonic limit, i.e. when the charging energy is small compared to the Josephson energy. The method allows one to derive an effective Hamiltonian for the capacitively shunted junction, starting from an arbitrary quadratic Hamiltonian describing the quasiparticles. The effective Hamiltonian takes the form of a generalized Jaynes-Cummings model describing the interaction between Josephson plasma modes and quasiparticle excitations. This model describes the energy spectra obtained from the MPS simulations deep in the harmonic limit quite well, but cannot reproduce non-perturbative effects that arise away from this limit (and are fully captured by the MPS simulations), such as the charge dispersion of energy levels and certain couplings between plasma modes and fermionic modes.

The paper is structured as follows. In Sec. 2 we present the setup and the general model used to describe a nanowire-based Josephson junction shunted by a capacitor, incorporating the fermionic degrees of freedom. In addition, we discuss the experimentally relevant probes and the parameter regimes we will be addressing in this study. As a simple application of the general model, in Sec. 3 we discuss and review a minimal model with a single low-energy fermionic mode on each side of the junction, which captures the essential ingredients of a nanowire in the topological phase. In Sec. 4 we discuss how MPS-based techniques can be used to calculate the experimentally relevant quantities that probe the response of the system. We then introduce the microscopic model for the nanowire that we use in our numerical study, and present the results. In Sec. 5 we consider the limit of small charging energy, and derive an effective theory based on a perturbative expansion which successfully captures the coupling between the plasma mode and the fermionic quasiparticles in this limit. We then discuss the effect of non-perturbative corrections.

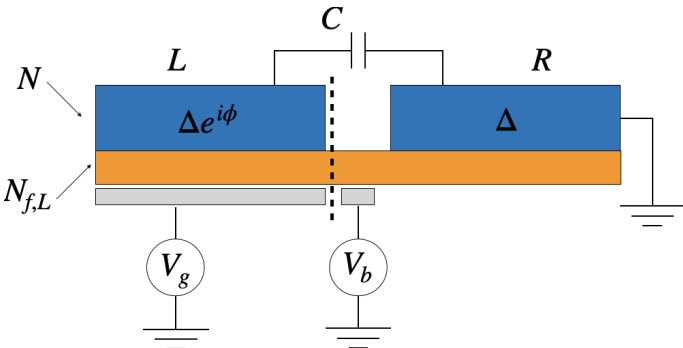

Figure 1: Schematic setup: a semiconducting nanowire proximity coupled to a superconductor, forming a Josephson junction shunted by a capacitor and controlled by the gate voltage $V_g$. The right half of the wire is connected to a superconducting ground. The dashed line represents the partitioning of the system used in our theoretical model, see main text for details.

## 2 Setup and model

The setup we consider is schematically depicted in Fig. 1. A semiconducting nanowire is proximity-coupled to a grounded superconductor on its right half, and to a floating superconducting island on its left half. A short segment in the middle of the wire, which is not in direct contact to any superconductor, forms a junction between the two superconductors. The conductance of the junction can be tuned by a gate underneath this middle region. The voltage on this gate is denoted by $V_b$ and determines the strength of the Josephson coupling between the floating superconducting island and the grounded superconductor. The island is shunted by a large capacitance $C$ to the ground. The charge induced on the island is controlled by a gate with voltage $V_g$.

The Hamiltonian describing the system is given by

$$H = E_c(N + N_{f,L} - N_g)^2 + \tfrac{1}{2}\mathbf{c}^\dagger H_{\text{BdG}}(\phi)\,\mathbf{c}\,. \tag{1}$$

The first term above is the electrostatic energy of the island, with charging energy $E_c = e^2/2C$ and dimensionless gate charge $N_g = CV_g/e$, where $e$ is the electron charge. The electrostatic energy is determined by the total number of electrons on the island (counted from the neutrality point), which is the sum of the number of paired electrons in the superconductor, $N$, and of the number of electrons in the left segment of the semiconducting wire, $N_{f,L}$ (to be better specified below).

The second term in Eq. (1) describes the dynamics of the fermionic degrees of freedom in the semiconducting wire. The dynamics are prescribed by a Bogoliubov-de Gennes (BdG) Hamiltonian, $H_{\text{BdG}}$, which includes the coupling between the semiconductor wire and the superconductors as well as the coupling between the two wire segments across the junction. For concreteness, we consider a lattice description of the system, such that $H_{\text{BdG}}$ is written in the Nambu basis

$$\mathbf{c}^\dagger = (c^\dagger_{i,\uparrow}, c^\dagger_{i,\downarrow}, c_{i,\uparrow}, c_{i,\downarrow})\,, \tag{2}$$

where $c^\dagger_{i,\sigma}$ ($c_{i,\sigma}$) is the creation (annihilation) operator of an electron on site $i$ with spin $\sigma$.

To simplify the treatment of the charging energy, we consider a sharp boundary between the left part of the wire, which is coupled to the floating superconducting island, and the right part of the wire, which is coupled to the grounded superconductor. We choose to place this sharp boundary at the left end of the junction, as indicated by the dashed black line in Fig. 1. Although this choice can have a quantitative effect on the spectrum of the full Hamiltonian

(with other parameters held fixed), we expect the qualitative physics to be insensitive to a specific (but generic) choice for the position of the boundary.

The sites $i$ in the lattice description of the wire are divided into two sets, $I_L$ and $I_R$, depending on whether they belong to the left or to the right part of the wire. The number of electrons in the left part of the wire is defined as

$$N_{f,L} = \sum_{i \in I_L} \left( c_{i,\uparrow}^\dagger c_{i,\uparrow} + c_{i,\downarrow}^\dagger c_{i,\downarrow} \right) . \tag{3}$$

The induced $s$-wave superconductivity is included in $H_{\mathrm{BdG}}$ via pairing terms of the form $\Delta e^{i\phi} c_{i,\uparrow} c_{i,\downarrow} + \text{h.c.}$ (if $i$ belongs to the part of the wire coupled to the floating island) or $\Delta c_{i,\uparrow} c_{i,\downarrow} + \text{h.c.}$ (if $i$ belongs to the part of the wire coupled to the grounded superconductor), where $\Delta$ is the induced superconducting gap. The pairing vanishes in the junction region. In what follows we will consider junctions of finite extent as well as junctions consisting of a single weak link. The operator $e^{i\phi}$ ($e^{-i\phi}$) adds (removes) a Cooper pair to (from) the left superconductor, and is canonically conjugate to the charge operator $N$, i.e.

$$[N, e^{\pm i\phi}] = \pm 2 e^{\pm i\phi} . \tag{4}$$

The pairing terms in $H_{\mathrm{BdG}}$ thus commute with the total charge of the floating island, $N + N_{f,L}$, which enters in the charging energy in Eq. (1). On the other hand, hopping terms in $H_{\mathrm{BdG}}$ which connect $I_L$ and $I_R$ do not commute with the charging energy term. More general forms of the induced pairing, e.g. a spatial variation of the pairing term strength or different pairing symmetries, could easily be included.

If the fermionic quasiparticles are gapped and one is interested in the behavior of the system at frequencies $\omega$ far below the excitation energy for quasiparticles, i.e. $\omega \ll \Delta$, then one may replace the Hamiltonian $H_{\mathrm{BdG}}$ with its phase-dependent ground state energy,

$$E_{\mathrm{GS}}(\phi) = -\frac{1}{2} \sum_n \epsilon_n(\phi) , \tag{5}$$

where $\epsilon_n(\phi)$ are the positive energy eigenvalues of $H_{\mathrm{BdG}}$ [1]. For a weakly transparent junction, such as the tunnel oxide junctions used in Al-based superconducting devices, the energy $E_{\mathrm{GS}}(\phi)$ is well-approximated by the form $-E_J \cos\phi$. In this case, one recovers the canonical superconducting qubit Hamiltonian, $H = E_c \left( N - N_g \right)^2 - E_J \cos(\phi)$. In the "transmon" limit $E_J \gg E_c$, the low-energy excitations of the junction are quantized charge oscillations—due to Cooper-pair tunneling across the junction—with a characteristic plasma frequency,

$$\omega_p = \sqrt{8 E_J E_c} , \tag{6}$$

and (crucially for qubit applications) a slightly anharmonic spectrum.

If, on the other hand, $H_{\mathrm{BdG}}$ has low-energy quasiparticle excitations with energies $\epsilon_n \lesssim \omega_p$, this description is no longer valid. Any correct description must include both the bosonic and the fermionic low-energy degrees of freedom present in the Hamiltonian of Eq. (1). Low-energy fermionic excitations naturally appear if the system is driven into a topological superconducting phase, where zero-energy Majorana modes emerge at the spatial boundaries between trivial and topological regions in the system. Moreover, as the system undergoes the topological phase transition between the conventional and the topological superconducting phase, the gap in the bulk of the system closes, giving rise to a continuum of states at energies below the plasma frequency. In general, one may expect sub-gap quasiparticles to arise

---

[1] More precisely, the sum runs over the eigenvalues $\epsilon_n(\phi)$ such that $\epsilon_n(0) > 0$, where the index $n$ must be assigned such that the associated BdG eigenfunctions vary smoothly as $\phi$ is varied.

quite generically in nanowire-based Josephson junctions as well as in junctions based on other systems engineered to support topological superconductivity, such as proximitized surfaces of topological insulators [77]. In these systems, many competing effects are involved, such as large magnetic fields, spin-orbit coupling, and interfaces between materials with very different properties, which can lead to complicated junction spectra even when the system is not tuned to the topological phase.

In a realistic model of a nanowire, the number of fermionic degrees of freedom appearing in $H_{\mathrm{BdG}}(\phi)$ may be too large to treat exactly. However, we expect that the effect of high-energy quasiparticles at energies $\epsilon_n \gg \omega_p$ can be captured by their contribution to the phase-dependent ground state energy. Assuming the latter can be approximated by a cosine dispersion, we rewrite the Hamiltonian in Eq. (1) as

$$H = E_c(N + N_{f,L} - N_g)^2 - E_J^0 \cos(\phi) + \tfrac{1}{2}\mathbf{c}^\dagger H_{\mathrm{BdG}}(\phi)\,\mathbf{c}\,, \tag{7}$$

where now the fermionic degrees of freedom $\mathbf{c}$ correspond to the low-energy degrees of freedom, and $E_J^0$ accounts for the high-energy degrees of freedom.

## 2.1 Experimental probes and parameters

The dynamics of the junction can be experimentally probed in a circuit quantum electrodynamics (cQED) setup [6], in which the junction is coupled to a microwave resonator cavity. The system is driven by an AC field, which corresponds to a time-dependent voltage $V_g$ in Fig. 1, $V_g(t) = V_g + \delta V_g(t)$. This time-dependent voltage couples to the total charge operator on the left island, $N_{\mathrm{tot}} = N + N_{f,L}$, leading to a small time-dependent contribution to the Hamiltonian, $\delta H(t) = E_c N_{\mathrm{tot}}(C/e)\,\delta V_g(t)$. We will characterize the response of the system to this perturbation by the spectral function of the charge operator,

$$S_N(\omega) = \int dt\, e^{-i\omega t}\langle 0|N_{\mathrm{tot}}(t)N_{\mathrm{tot}}(0)|0\rangle = \sum_\alpha \delta(\omega - (E_\alpha - E_0))\,|\langle \alpha|N_{\mathrm{tot}}|0\rangle|^2\,. \tag{8}$$

Here, $|\alpha\rangle$ denotes an eigenstate of the system with energy $E_\alpha$, with $\alpha = 0$ the ground state. The spectral function exhibits a peak at each transition energy of the system, with the intensity of the peak related to the matrix element of the total charge operator between the initial and final states of the transition.

We now discuss interesting parameter regimes for typical cQED circuits which we consider in our simulations. Transmon qubits typically operate in the regime $E_J/E_c \approx 20$ or higher in order to suppress charge noise. $E_c$ typically varies in the 200–500 MHz range, yielding plasma frequencies in the 5–10 GHz range. Gate-controlled nanowire junctions allow for a tunable $E_J$ and thus allow a device to be operated not only in the transmon regime but also in a regime with lower $E_J/E_c$. The lower the ratio $E_J/E_c$, the larger the charge dispersion, i.e. the dependence of the energy levels on the dimensionless charge $N_g$. An intermediate ratio $E_J/E_c \approx 5$ is interesting since, as shown in Refs. [60,61], the presence of Majorana zero modes coupled across the junction is associated with distinct features in the charge dispersion.

## 2.2 Gauge transformation

To study the model (1), it is useful to perform a unitary gauge transformation, $H \mapsto UHU^\dagger$, with [78]

$$U = e^{i\phi N_{f,L}/2}. \tag{9}$$

Under this gauge transformation, the number operator $N = -2i\partial_\phi$ transforms as $N \mapsto N - N_{f,L}$. This simplifies the first term in Eq. (1), which after the transformation is given by $E_c(N - N_g)^2$.

In addition, the fermion operators that belong to the left island transform as $c_{i,\sigma} \mapsto e^{-i\phi/2}c_{i,\sigma}$, while the fermions on the right island are unaffected by the transformation. Hence, after the transformation, the operators $c_{i,\sigma}$ with $i \in I_L$ are charge-neutral, while the operator $N$ counts the total charge of the superconducting island *and* of the left part of the wire.

The effect of the transformation on the terms in $H_{\mathrm{BdG}}(\phi)$ is the following. The pairing terms on the left part of the wire, which initially take the form $\Delta e^{i\phi}c_{i,\uparrow}c_{i,\downarrow}$, lose their phase dependence and are given by $\Delta c_{i,\uparrow}c_{i,\downarrow}$. Meanwhile, terms describing hopping between the left and the right parts of the wire acquire phase dependence: $c_{i,\sigma}^{\dagger}c_{j,\sigma'}$ with $i \in I_L$ and $j \in I_R$ becomes $e^{i\phi/2}c_{i,\sigma}^{\dagger}c_{j,\sigma'}$.

We must also discuss the effect of the gauge transformation on the wave functions. A complete basis for the Hilbert space of the model (1) is given by $|n, \vec{n}_L, \vec{n}_R\rangle$, where $\vec{n}_{L(R)}$ denotes the vector of occupation numbers for the fermionic degrees of freedom on the left (right) part of the wire, and $n \in \mathbb{Z}$ is the number of *Cooper pairs* (counted from charge-neutrality) in the left superconductor, i.e. $N|n, \vec{n}_L, \vec{n}_R\rangle = 2n |n, \vec{n}_L, \vec{n}_R\rangle$. A generic many-body state is thus given (in the original gauge) as

$$|\Psi\rangle = \sum_{n,\vec{n}_L,\vec{n}_R} \Psi(n,\vec{n}_L,\vec{n}_R) |n,\vec{n}_L,\vec{n}_R\rangle . \tag{10}$$

Although we will eventually use this "number basis" in the numerics, it is convenient to temporarily work in the "phase basis", formed by the states

$$|\phi,\vec{n}_L,\vec{n}_R\rangle = (2\pi)^{-1} \sum_n e^{i\phi n} |n,\vec{n}_L,\vec{n}_R\rangle . \tag{11}$$

The wave function coefficients in the phase basis are

$$\Psi(\phi,\vec{n}_L,\vec{n}_R) = \sum_n e^{-i\phi n}\Psi(n,\vec{n}_L,\vec{n}_R) . \tag{12}$$

They are $2\pi$-periodic: $\Psi(\phi + 2\pi,\vec{n}_L,\vec{n}_R) = \Psi(\phi,\vec{n}_L,\vec{n}_R)$. The action of the gauge operator $U$ on a phase basis state is simply

$$U|\phi,\vec{n}_L,\vec{n}_R\rangle = e^{i\phi n_L/2}|\phi,\vec{n}_L,\vec{n}_R\rangle , \tag{13}$$

where $n_L$ is the number of electron in the left island. Thus, the gauge transformation maps the state $|\Psi\rangle$ to a new state, $|\tilde{\Psi}\rangle = U|\Psi\rangle$, with wavefunction coefficients

$$\tilde{\Psi}(\phi,\vec{n}_L,\vec{n}_R) = e^{i\phi n_L/2}\Psi(\phi,\vec{n}_L,\vec{n}_R) . \tag{14}$$

Because of the extra phase factor, the boundary conditions for $\tilde{\Psi}$ are periodic or anti-periodic depending on the parity of the number of fermions in the left island [79]:

$$\tilde{\Psi}(\phi + 2\pi,\vec{n}_L,\vec{n}_R) = e^{i\pi n_L}\tilde{\Psi}(\phi,\vec{n}_L,\vec{n}_R) . \tag{15}$$

With this new boundary condition, the spectrum of the operator $N = -2i\partial_\phi$ changes from $2\mathbb{Z}$ (in the original gauge) to $\mathbb{Z}$ (in the new gauge). This is in agreement with the fact that, as mentioned above, $N$ must now account for the *total* charge of the left island and not only for the part due to the paired electrons. However, this accounting is only consistent if the parity of $N$, which we refer to as the bosonic parity and denote by $P_b = e^{i\pi N}$, is the same as the parity of the number of (now charge-neutral) fermions on the left island, $P_{f,L} = e^{i\pi N_{f,L}}$. That is, in the new gauge every physical state must obey the following constraint:

$$P_b|\tilde{\Psi}\rangle = P_{f,L}|\tilde{\Psi}\rangle . \tag{16}$$

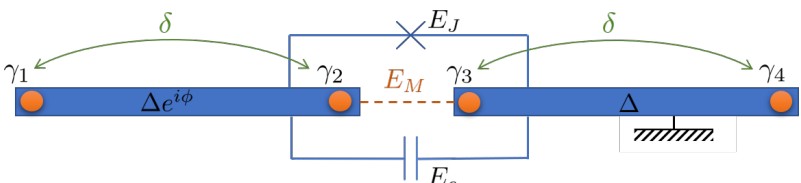

Figure 2: Minimal model describing a nanowire in the topological phase, with Majorana zero modes, $\gamma_{i=1,..,4}$, at the ends of the topological superconducting regions.

Note that, since $P_b = e^{i\pi N}$ is the operator that translates $\phi$ by $2\pi$, the constraint (16) is simply a rewriting of (15) in a basis-independent form.

Finally, we comment on the effect of the gauge transformation on the calculation of the spectral function (8). Since, after the gauge transformation, $N$ is the *total* charge on the left island, the correlation function that appears in the integral in Eq. (8) is now simply $\langle N(t)N(0)\rangle$.

## 3 Minimal model for a nanowire in the topological superconducting phase

We now turn to a minimal realization of Eq. (7), namely the case in which there is exactly one low-energy fermionic mode on each side of the junction. The setup is chosen to capture the essential ingredients of a nanowire in the topological phase, hosting one Majorana mode at each end of each topological segment. We denote the Majorana modes at the ends of the left (right) island by $\gamma_{1,2}$ ($\gamma_{3,4}$) as depicted in Fig. 2. Assuming the absence of additional low-energy fermionic quasiparticles, the effective Hamiltonian, written in the gauge where the boundary conditions of Eq. (15) hold, is given by:

$$H = H_J + H_\delta + H_M \, , \tag{17}$$

with

$$H_J = E_c(N - N_g)^2 - E_J \cos\phi \, , \tag{18a}$$
$$H_\delta = \delta(i\gamma_1\gamma_2 + i\gamma_3\gamma_4) \, , \tag{18b}$$
$$H_M = i\gamma_2\gamma_3 E_M \cos(\phi/2) \, . \tag{18c}$$

The term $H_\delta$ couples the Majorana modes within each island. Such a coupling can arise due to the finite length of each island, and it is chosen to be identical on both islands for simplicity. In the topological phase, $\delta$ vanishes exponentially with the length of each island. The phase-dependent coupling $E_M \cos(\phi/2)$ originates from single-electron tunneling across the junction. This model was introduced and studied by Ginossar and Grosfeld [60]. In the present work we discuss it in the context of the more general model (1), and address additional points that were not discussed in detail in Ref. [60].

To solve the model (17), we first deal with $H_J$ and $H_\delta$ separately, and then consider the effect of $H_M$, which couples the bosonic and fermionic excitations. The eigenfunctions of $H_J$ are known exactly in terms of Mathieu functions (approximate eigenfunctions are also immediate to find numerically). Anticipating the role of the boundary conditions (15), we will find eigenfunctions in the space of $4\pi$-periodic functions. The $2\pi$-periodicity of $H_J$ implies that $[H_J, P_b] = 0$, and hence we can choose eigenstates to have well-defined bosonic parity. We denote the $m$-th energy eigenstate with even/odd bosonic parity by $|m, \pm\rangle_b$, and the

corresponding energy by $E_{m,\pm}$. The wave functions $\psi_{m,\pm}(\phi) = \langle \phi | m, \pm \rangle_b$ are given by

$$\psi_{m,\pm}(\phi) = \frac{e^{i(\phi N_g + \pi \nu_{m,\pm})/2}}{\sqrt{4\pi}\, i^m} \mathrm{me}_{\nu_{m,\pm}}\left(\frac{\phi - \pi}{2}, \frac{E_J}{2E_c}\right) , \qquad (19)$$

where $\mathrm{me}_\nu(z, q)$ is the Mathieu function with characteristic exponent $\nu$, and where

$$\nu_{m,\pm} = \tfrac{1}{2} \mp (-1)^m (m + \tfrac{1}{2}) - N_g . \qquad (20)$$

The wave functions satisfy the boundary condition $\psi_{m,\pm}(\phi + 2\pi) = \pm\psi_{m,\pm}(\phi)$. In the limit $E_J \ll E_c$, they reduce to normalized plane waves, $\psi_{m,\pm}(\phi) \sim e^{ik_{m,\pm}\phi}/\sqrt{4\pi}$, with $k_{m,+} \in \mathbb{Z}$ and $2k_{m,-} \in \mathbb{Z}$. In the opposite limit $E_J \gg E_c$, the wavefunctions are localized near the minima of the potential $-E_J \cos \phi$, i.e. near $\phi = 0$ and $2\pi$.

To describe the fermionic sector, we observe that two Majorana modes together form a fermionic mode, and define its occupation as $n_{ij} = (1 + i\gamma_i\gamma_j)/2$. A basis for the fermionic Hilbert space can thus be obtained by arranging the Majorana modes into pairs and specifying the occupations of the pairs. We focus on the sector with even total fermion parity, and take the pairs of Majorana modes to be those in the same superconducting region, such that the basis states also have well-defined fermion parity in each island. Then, the two possible states are $|n_{12} = n_{34} = 0\rangle$ and $|n_{12} = n_{34} = 1\rangle$.

We now define a basis for the full Hilbert space describing both the bosonic and the fermionic sectors, using states which obey the constraint in Eq. (16):

$$|m, +\rangle = |m, +\rangle_b \otimes |n_{12} = n_{34} = 0\rangle , \qquad (21a)$$

$$|m, -\rangle = |m, -\rangle_b \otimes |n_{12} = n_{34} = 1\rangle . \qquad (21b)$$

These states have equal boson parity and fermion parity on each island; for the remainder of this section, we will refer to this as island parity. The Hamiltonian $H_J + H_\delta$ is diagonal in this basis; the state $|m, p\rangle$ has energy $E_{m,p} - 2p\delta$, where $p = \pm 1$ is the island parity. The four lowest-energy states in the regime with $\delta \ll \omega_p$ ($m = 0, 1$ and $p = \pm$) are shown schematically in Fig. 3(a), illustrating the parity constraint.

The coupling term $H_M$ is entirely off-diagonal in this basis, since it couples states of opposite island parity. Its nonzero matrix elements are

$$\langle m, + | H_M | m', - \rangle = -\eta_{m,m'} E_M , \qquad (22)$$

$$\eta_{m,m'} = \int_0^{4\pi} d\phi \, \psi_{m,+}^*(\phi) \cos(\phi/2) \psi_{m',-}(\phi) . \qquad (23)$$

In the limit $E_J \gg E_c$, an asymptotic analysis of the integral in (23) shows that the dominant matrix elements are the diagonal ones, $\eta_{m,m} = 1 - O(z)$, where $z = \sqrt{2E_c/E_J} \ll 1$. The matrix elements in which $m$ and $m'$ differ by an even integer $2\ell$ are subdominant, $\eta_{m-\ell,m+\ell} = O(z^\ell)$. The matrix elements in which $m$ and $m'$ differ by an odd integer, $\eta_{m,m+2\ell+1}$, are exponentially small in the ratio $E_J/E_c$. They vanish identically when $N_g$ is an integer and are largest when $N_g$ is half-integer.

The low-energy spectrum as a function of $N_g$ is shown in Fig. 3(b) in the two limits of large $\delta \gg \omega_p$ (left panel) and $\delta = 0$ (right panel). The colored dashed lines correspond to the spectrum for $E_M = 0$, while the solid black lines correspond to the spectrum in the presence of a finite $E_M$. The dispersion of energy with $N_g$ can be understood in terms of instanton tunneling processes between the two minima of the potential at $\phi = 0$ and $2\pi$ (quantum phase slips), and has a magnitude proportional to $e^{-\sqrt{8E_J/E_c}}$. The dispersion of levels with opposite island parity is shifted by one unit along $N_g$, leading to level crossings at half-integer values of $N_g$

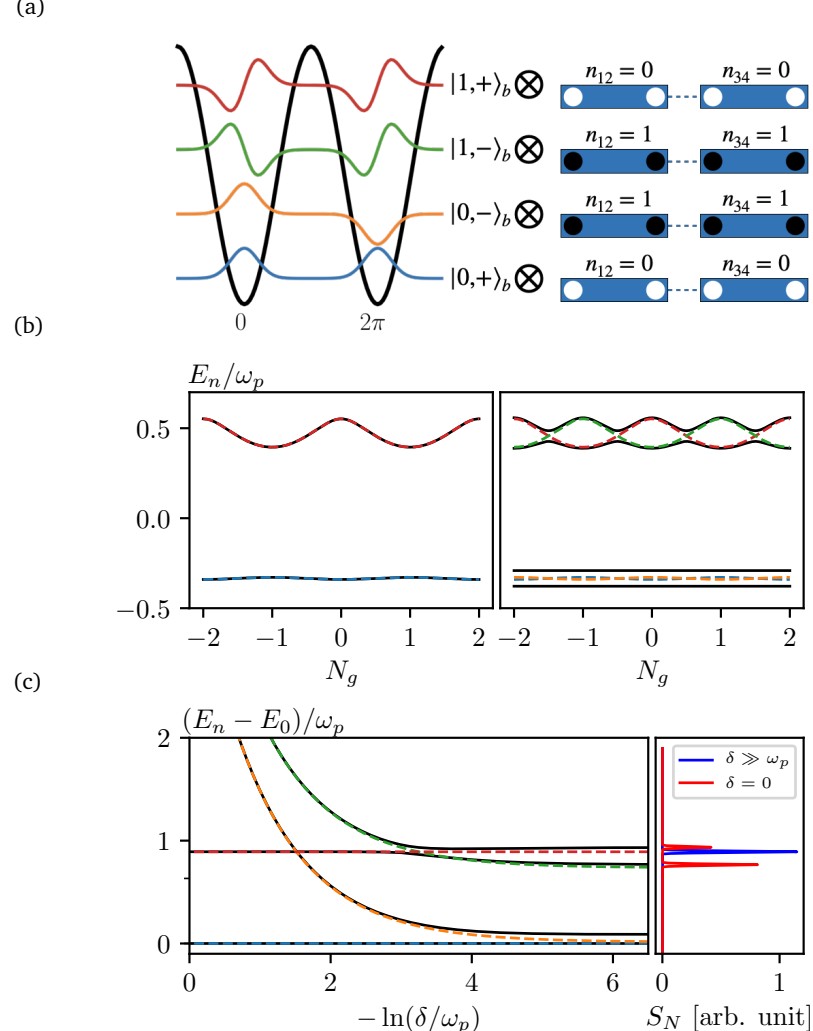

**Figure 3:** (a) Schematic representation of the basis states defined in Eq. (21) for $m = 0, 1$, in terms of the $4\pi$-periodic bosonic wavefunctions $\psi_{m,\pm}(\phi)$, and the occupations of the four Majorana modes present in the system. (b) The energy spectrum of the Hamiltonian (17), as a function of $N_g$, for large $\delta \gtrsim \omega_p$ (left panel) and for $\delta = 0$ (right panel), with junction parameters set to $E_J/E_c = 5$. The colored dashed lines correspond to the spectrum for $E_M = 0$, where the eigenstates are precisely the states sketched in panel (a), while the solid black lines are obtained for $E_M/E_c = 0.3$. (c) Excitation spectrum as a function of $\delta/\omega_p$ for $N_g = 0$, and the spectral function at $\delta = 5\omega_p$ (blue) and $\delta = 0$ (red). The spectral function is convolved with a Gaussian with $\sigma/\omega_p = 10^{-2}$.

at $\delta = 0$, see right panel of Fig. 3b. $H_M$ introduces a coupling proportional to $E_M \eta_{m,m}(N_g)$ between states of oppposite parity, leading to avoided crossings at the degeneracy points.

We now study the behavior of the excitation spectrum for a choice of parameters that mimics driving the system from a trivial and fully gapped superconducting phase into a topological phase with well-separated Majorana zero modes. Fixing $N_g = 0$ and $E_J/E_c = 5$, we vary the coupling between the Majorana modes on the same island from a large value $\delta \gg \omega_p$ to a small value $\delta \ll \omega_p$, and compare the excitation spectrum with $E_M = 0$ to the one with a small but finite $E_M \ll \omega_p$. The resulting spectra are shown in Fig. 3(c).

For large $\delta \gg \omega_p$, the ground state is $|0, +\rangle$ and all the states with odd island parity, $|m, -\rangle$, are at high energy. The spectrum of the junction resembles that of a trivial Josephson

junction, with level spacing set by $\omega_p$, up to anharmonic corrections. When $\delta \approx \omega_p/4$, there are degeneracies in the spectrum between states $|m,-\rangle$ and $|m+1,+\rangle$. At finite $E_M$ the coupling between these states is proportional to $\eta_{m+1,m}$, and hence vanishes for $N_g = 0$. For $\delta \ll \omega_p$, the two states in each doublet, $|m,\pm\rangle$, get closer in energy. In the limit $\delta = 0$ the energy splitting between these states is set by the larger of the two energy scales $E_M$ and the splitting due to charge dispersion, $|E_{m,+} - E_{m,-}|$. Note that for the doublets with odd $m$, one has $E_{m,-} < E_{m,+}$ at $N_g = 0$, and thus the state with odd island parity crosses the one with even parity in energy as $\delta$ is decreased. In Fig. 3(c) this can be seen as the crossing of the lines corresponding to $|1,+\rangle$ (red) and $|1,-\rangle$ (green). At finite $E_M$, the coupling between these states gives rise to an avoided crossing between them, with size of order $\eta_{m,m}E_M$.

Using the excitation spectrum, we calculate the spectral function (8) for $\delta \gg \omega_p$ and $\delta = 0$ (see right panel of Fig. 3(c)). When $\delta \gg \omega_p$, the ground state is simply $|0,+\rangle$. Since the operator $N$ can only couple states with the same parity, only the spectral lines within the even parity sector are observed in the spectral response. Once different parity states couple due to finite $E_M$, additional spectral lines can be observed. In particular, for $\delta = 0$, both spectral lines in the $m = 1$ manifold can be observed. Deep in the harmonic limit, when $|E_{m,+} - E_{m,-}| \to 0$, the eigenstates are superpositions of states with well-defined fermionic parity of the junction (i.e. occupation of $n_{23}$), and due to the total fermionic parity constraint also of $n_{14}$. As the charge operator $N$ acts locally at the junction, it cannot change the occupation of $n_{14}$, and only a single spectral line is visible again. For a further discussion of this limit, see Sec. 5.2.

## 4 Numerical simulations

Building on the picture developed in the previous sections, we now turn to a numerical study, using DMRG and TEBD, of the spectral function defined in Eq. (8). We will perform the study on a microscopic model that allows us to treat the behavior both in the topological phase and in the vicinity of the topological phase transition, as well as to examine the effect of additional Andreev subgap states in the junction both in the topological and non-topological regime.

Numerical simulations presented in this work were performed using the ITensor library [80].

### 4.1 Lattice model and numerical techniques

We model the system as a spinful wire with Rashba spin-orbit coupling that is proximity-coupled to an $s$-wave superconductor. A large enough Zeeman field perpendicular to the direction of the spin-orbit coupling drives the system into the topological superconducting phase [81,82]. To simplify the numerics, we will consider a single spinful band, while additional modes that can be present in a real system will be accounted for by the term $-E_J^0 \cos\phi$ in the Hamiltonian (7). Note that the band which is considered explicitly will also contribute to the Josephson coupling across the junction due to its ground state energy dispersion with phase, which (in the limit of small transmission through the junction) can be approximated by $-E_J^1 \cos(\phi)$. The total Josephson coupling is then given by $E_J = E_J^0 + E_J^1$. In practice, for parameters discussed in the rest of this section $E_J^1 \ll E_J^0$, and hence $E_J \approx E_J^0$.

The continuum version of the normal part of the BdG Hamiltonian is given by

$$H_0 = \frac{k_x^2}{2m} - \mu + \alpha k_x \sigma_y + B \sigma_x . \tag{24}$$

In our numerical simulations we use a lattice description of the system. To this end, we introduce the tight-binding parameters corresponding to the hopping amplitude, $t = 1/(2ma^2)$, and the spin-orbit coupling, $\nu = \alpha/a$, where $a$ is the lattice constant (see Appendix A for

the explicit tight-binding Hamiltonian). Unless otherwise specified, all energies hereafter are measured in units of $\Delta$.

We describe the system in the gauge introduced in Sec. 2.2, where the Hilbert space consists of a lattice of fermionic degrees of freedom and a bosonic degree of freedom describing the total charge on the floating island. We represent this bosonic mode in the charge basis and as a single additional site. The occupation of this charge mode depends on the ratio $E_J/E_c$, but the number fluctuations grow slowly, $\langle \delta N^2 \rangle \sim (E_J/E_c)^{1/2}$ for $E_J > E_c$ [4], so in practice we find that truncating to $10 - 20$ charge states yields sufficiently small error. The charging energy term acts only on the bosonic charge state, which is coupled to the fermionic modes via the hopping terms between the left and the right parts of the wire, i.e. via $e^{i\phi/2} c_{i,\sigma}^\dagger c_{j,\sigma'}$ with $i \in I_L$, $j \in I_R$ (recall that in the charge basis the operator $e^{i\phi/2}$ simply changes the charge by one). Hence, if the charge site is placed at the boundary between the left and the right part of the wire, the Hamiltonian is local, simplifying the numerical study. The constraint introduced by the boundary conditions (15) is handled by defining a conserved $\mathbb{Z}_2$ quantum number $P_b P_{f,L} = 1$.

To obtain the spectral function $S_N(\omega)$ as defined in Eq. (8) we first obtain the ground state $|0\rangle$ of the system using DMRG. We then perform time evolution of the state $|\tilde{0}\rangle \equiv N|0\rangle$ in order to obtain the real-time correlation function $\langle 0|N(t)N(0)|0\rangle = e^{-iE_0 t}\langle \tilde{0}|e^{iHt}|\tilde{0}\rangle$. For the time evolution, we use TEBD with a 4th order Suzuki-Trotter decomposition with a time step of $dt = 0.02$, truncating the MPS wavefunction to bond dimension of $M_{\max} = 50$ and truncation error $\epsilon_{\text{tr}} = 10^{-7}$. We note that similar numerical techniques could be used to characterize the correlation function for some other initial state, such as a low-lying excited state or even a mixed state. This may be of interest to describe experimental situations where finite temperature or non-equilibrium effects lead to a finite occupation of excited states. The computational cost of this time evolution is heavily dominated by the terms of the Hamiltonian that involve the bosonic degree of freedom, which has a large on-site Hilbert space dimension and thus limits the bond dimension we can treat. To obtain the spectral function in real frequencies, we typically compute the real-time correlation function up to times $t_f \simeq 400$ and use linear prediction methods [83–85] to extrapolate the real-time correlation function to times $\sim 2t_f$. We then apply a Gaussian windowing function and finally perform the Fourier transformation. This allows us to reach a frequency resolution of order $10^{-2}\omega_p$ in the parameter regime we consider.

We consider two different models for the junction, corresponding to the limits of a short and a finite-length junction, that will be described in detail below.

## 4.2 Short junction (weak-link model)

We start from the short junction limit, valid for junctions much shorter than the superconducting coherence length, in which the junction can be modeled as a single weak link between the islands. All hopping terms on this link are reduced by a factor $\kappa < 1$ compared to their values in the bulk. The transmission of the junction, and hence $E_M$, is determined by $\kappa$; for small $\kappa$, $E_M$ is proportional to $\kappa$. This setup is shown schematically in Fig. 4(a). The exact tight-binding Hamiltonian is given in Appendix A.

In Fig. 4(b), we plot the spectral function obtained for this model, as a function of the Zeeman energy $B$, for $N_g = 0$ and $E_J^0/E_c = 5$. In the trivial phase, for $B \lesssim B_c$, a single spectral line is present (in the frequency range shown) at a frequency $\omega \approx \omega_p$, as expected. As the Zeeman energy is increased and the wire is driven into the topological phase, a second spectral line appears due to finite $E_M$ as discussed in Sec. 3. The avoided crossing between the two spectral lines in Fig. 4(b), which can be observed close to the topological phase transition at $B_c/\Delta = 1$, is reminiscent of the avoided crossing between the states $|1,+\rangle$ and $|1,-\rangle$ in

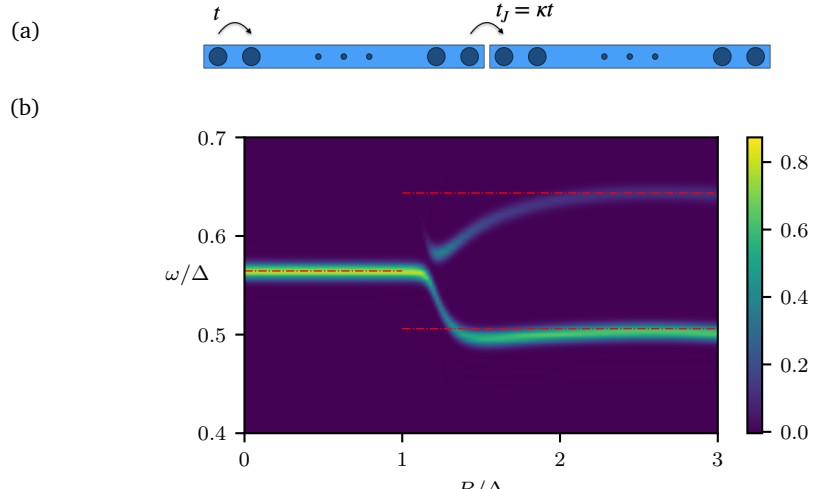

Figure 4: (a) Short junction (weak-link) model. The junction comprises a single link on which the hopping and the spin-orbit coupling are reduced by a factor $\kappa < 1$ compared to their values in the bulk. (b) Spectral function obtained with DMRG and TEBD for the weak-link model, as a function of the Zeeman energy $B$ for $N_g = 0$. Junction parameters in units of $\Delta$ are $E_c = 0.1$, $E_J^0 = 0.5$, and wire tight-binding parameters are $t = 1.5$, $\nu = 2$, $\mu = 0$, $\kappa = 0.1$. The length of each part of the wire is $N_x = 40$. In the trivial phase, for $B/\Delta \lesssim B_c/\Delta = 1$, a single spectral line is seen, corresponding to the plasma frequency. As the system enters the topological phase, a second spectral line appears, and an avoided crossing between the two can be observed (see main text for further discussion). The red dashed lines correspond, for $B < B_c$, to the spectral lines expected for a conventional Josephson junction, and for $B > B_c$, to those expected for a junction deep in the topological phase. The latter are obtained using the minimal model given by the Hamiltonian (17) (see main text for more details).

Fig. 3(b).

To validate our numerical results, we overlay on the spectral function the spectral lines expected deep in the trivial and the topological phase, obtained using the minimal model discussed in Sec. 3. These are plotted as red dashed lines in Fig. 4(b). For the trivial phase ($B < B_c$), the position of the spectral line is calculated from $H_J$ of Eq. (18a) with Josephson energy set to $E_J^0$. For the topological phase ($B > B_c$), we use the the Hamiltonian (17) with $\delta = 0$ and a value of $E_M$ determined numerically from the $4\pi$-periodic component of the ground state energy deep in the topological phase (more specifically, at $B/\Delta = 2$), as explained in detail in Appendix B. As can be seen, our numerical results indeed agree very well with these values in both limits.

### 4.2.1 Topological phase transition

The finite-size gap at the transition is determined by the spin-orbit coupling strength as $\epsilon \sim \alpha\Delta/(BL)$ (see Ref. [86] for details). For the parameters used to obtain Fig. 4, we have $\epsilon \sim \omega_p$. To probe the response of the system closer to the continuum limit we consider a smaller spin-orbit coupling, such that many states cross the plasma frequency close to the topological phase transition, but still large enough to have a sizable gap in the topological region. However, we still do not observe any significant features in the spectral response associated with the gap closing, as can be seen in Fig. 5, where we also plot the energies of

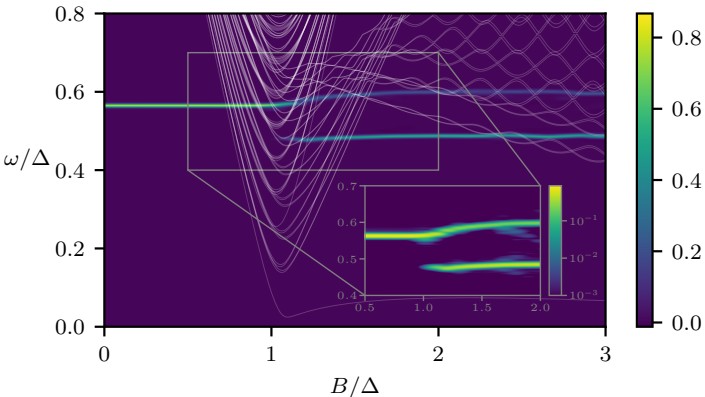

Figure 5: Spectral function obtained with DMRG and TEBD, for the weak-link model, as a function of the Zeeman energy $B$ for $N_g = 0$. Junction parameters in units of $\Delta$ are $E_c = 0.1$, $E_J^0 = 0.5$, and wire tight-binding parameters are $t = 1.5$, $v = 0.6$, $\mu = 0$, $\kappa = 0.1$. The length of each part of the wire is $N_x = 40$. White solid lines are the energies of 2-quasiparticle excitations obtained from the BdG Hamiltonian for a phase difference of $\phi = 0$ between the superconducting islands. Around the phase transition point $B_c/\Delta = 1$, as the bulk gap is closing, multiple 2-quasiparticle states cross the plasma frequency. However, the effect of these states on the spectral function is very small. The inset shows a zoom-in on the region close to the phase transition, on a logarithmic color scale.

the two-quasiparticle excitations on top of the spectral function. We attribute this to the fact that the critical states have weak dependence on the phase difference at the junction, and thus small matrix elements of the charge operator $N$ which determines the spectral response via Eq. (8). An intuitive picture for this observation is that the critical states are delocalized throughout the entire wire length, and thus have a limited support close to the junction.

## 4.3 Finite-length junction

We now consider a finite-length junction, which is modeled as a finite segment of length $W$ of normal (non-superconducting) wire. The hopping and the spin-orbit coupling between the left (right) superconducting region of the wire and the junction is reduced by a factor $\kappa_{L(R)}$ compared to their values in the bulk. The exact tight-binding description of the model is given in Appendix A and is shown schematically in Fig. 6(a).

In Fig. 6(b) we plot the spectral function obtained for this model. We find that in this case the structure of the spectral function is more complicated with additional spectral lines appearing both in the trivial and in the topological phase. To understand this spectral function, we first obtain the spectrum expected on the basis of the minimal model of Sec. 3 deep in the trivial and topological phase, as we did for the short junction case (see previous subsection for more details). The red dashed lines plotted on top of the spectral function for $B < B_c$ ($B > B_c$) correspond to this spectrum. In addition, we plot as white solid lines the energies of two-quasiparticle excitations, obtained from the tight-binding BdG Hamiltonian at a phase difference of $\phi = 0$ between the superconducting islands, as a function of $B$. These are the fermionic excitations (in the even parity sector) originating from the second term in the Hamiltonian (1), neglecting the dynamics of the field $\phi$ and the finite charging energy. It can be clearly seen that the lines in the spectral response originate from avoided crossings between these two types of excitations. In Sec. 5 below, we will present a perturbative approach that

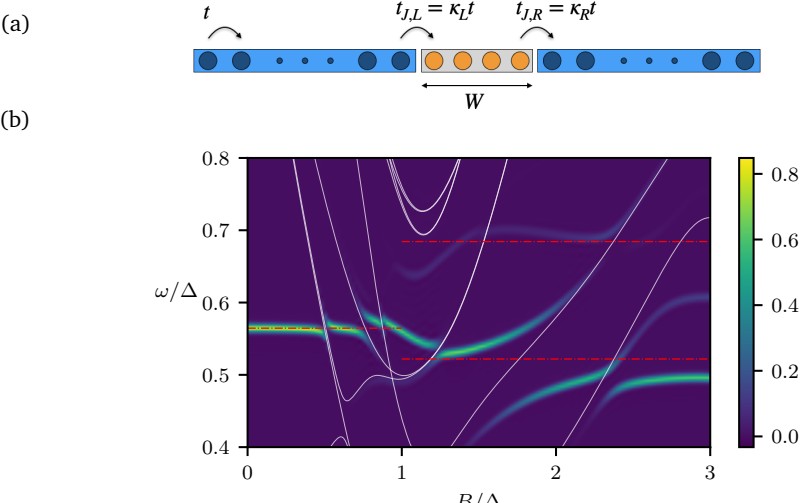

Figure 6: (a) Finite-length junction model. Sites depicted in blue correspond to the two parts of the nanowire which are proximitized. The orange sites in the middle region are not subject to a superconducting pairing term, but otherwise have the same model parameters as the rest of the wire. The hopping and the spin-orbit coupling between the proximity-coupled left (right) part of the wire and the normal junction region are reduced by a factor $\kappa_{L(R)}$ compared to their values in the bulk. (b) The spectral function for a finite-length junction model as a function of the Zeeman energy $B$ for $N_g = 0$. Junction parameters are $E_c = 0.1$, $E_J^0 = 0.5$, $N_g = 0$, and wire tight-binding parameters are $t = 1.5$, $v = 2$, $\mu = 0$, $\kappa_L = \kappa_R = 0.3$. The length of each part of the wire that is coupled to a superconductor is $N_x = 30$, and the length of the junction is $W = 6$. Red dashed lines, plotted on top of the spectral function for $B < B_c$ ($B > B_c$), are the spectral lines expected deep in the trivial (topological) phase (see main text for more details). White solid lines are the energies of 2-quasiparticle excitations obtained from the BdG Hamiltonian for a phase difference of $\phi = 0$ between the superconducting islands.

will allow us to understand this coupling and to calculate the avoided crossings in the harmonic limit, i.e. for $E_J/E_c \to \infty$.

# 5 Effective theory for the coupling between bosonic and fermionic modes

We now examine the observed avoided crossing betweens the bosonic plasma mode and fermionic subgap states in more detail. To this end, in Sec. 5.1 we develop a perturbative theory in the harmonic limit. In Sec. 5.2 we discuss non-perturbative couplings that cannot be captured by the perturbative expansion, and show their effect on the spectrum of the problem using a simple model.

## 5.1 Harmonic expansion

We start from the Hamiltonian (7), where we assumed that the effect of the high-energy quasiparticles is captured by their contribution to the phase dependent ground state energy that can be approximated by a cosine dispersion $-E_J^0 \cos(\phi)$. In the limit $E_J^0 \gg E_c$, phase fluctuations

are small and centered around $\phi = 0$. Therefore, we can ignore the fact that $\phi$ is a compact variable, and thus also the periodic or anti-periodic boundary conditions (15). In this approximation, the $N_g$-dependence of the Hamiltonian can be "gauged away" by generalizing the gauge transformation (9) to $U = e^{i\phi(N_{f,L} - N_g)/2}$ and the dependence of the eigenvalues on $N_g$ is lost.

Expanding the cosine dispersion to lowest order in $\phi$, we denote by $H_{0,b}$ the resulting harmonic oscillator Hamiltonian acting on the bosonic degree of freedom,

$$H_{0,b} = E_c N^2 + \tfrac{1}{2} E_J^0 \phi^2 = \omega_p^0 \left( a^\dagger a + \tfrac{1}{2} \right) . \tag{25}$$

Here $\omega_p^0 = \sqrt{8 E_J^0 E_c}$, and $a^\dagger$ and $a$ are harmonic oscillator raising and lowering operators, respectively. We will denote the eigenstates of $H_{0,b}$ as $|m\rangle$, where $m$ is the occupation level of the harmonic oscillator.

We now expand $H_{\text{BdG}}(\phi)$ around $\phi = 0$,

$$\mathbf{c}^\dagger H_{\text{BdG}}(\phi)\mathbf{c} = \mathbf{c}^\dagger \left[ H_{\text{BdG}}(0) + \phi\, H'_{\text{BdG}}(0) + \cdots \right] \mathbf{c} . \tag{26}$$

We denote the zeroth order term in this expansion by $H_{0,f}$ and diagonalize it to obtain

$$H_{0,f} = \tfrac{1}{2} \mathbf{c}^\dagger H_{\text{BdG}}(0)\, \mathbf{c} = \sum_i \epsilon_i \left( \Gamma_i^\dagger \Gamma_i - \tfrac{1}{2} \right) . \tag{27}$$

Here, the operators $\Gamma_i^\dagger$ ($\Gamma_i$) create (annihilate) quasiparticle excitations with non-negative energies $\epsilon_i$. We define a corresponding Nambu spinor $\mathbf{\Gamma}^\dagger = (\Gamma_i^\dagger, \Gamma_i)$ and denote the single-particle unitary that diagonalizes $H_{\text{BdG}}(0)$ by $V$, i.e. $\mathbf{c} = V\mathbf{\Gamma}$. We will denote the eigenstates of $H_{0,f}$ as $|\vec{v}\rangle$, where $\vec{v} = (v_i)$, with $v_i \in \{0,1\}$, is the vector of occupations of the quasiparticle levels. In particular, the ground state will be denoted as $|\vec{0}\rangle$. The excited states are then given explicitly as $|\vec{v}\rangle = \prod_i (\Gamma_i^\dagger)^{v_i} |\vec{0}\rangle$.

We take the sum $H_0 = H_{0,b} + H_{0,f}$ as the unperturbed Hamiltonian for the problem. Note that since $H_{0,b(f)}$ acts solely on the bosonic (fermionic) degrees of freedom, the eigenstates of $H_0$ are simply the tensor products $|m, \vec{v}\rangle \equiv |m\rangle \otimes |\vec{v}\rangle$. Their unperturbed energies are given by

$$E_{m,\vec{v}} = m\omega_p^0 + \sum_i v_i \epsilon_i + \text{const.} \tag{28}$$

The second term in the expansion (26) acts as a perturbation to $H_0$, which we denote as $\delta H$. This term introduces a coupling between the bosonic and fermionic degrees of freedom. Noting that the phase $\phi$ has the representation $\phi = \sqrt{z}(a + a^\dagger)$, where $z = (2 E_c / E_J^0)^{1/2}$ is a small parameter, and using the basis $|\vec{v}\rangle$ for the fermionic states, $\delta H$ can be written as

$$\delta H = \tfrac{1}{2} \sqrt{z}(a + a^\dagger)\, \mathbf{\Gamma}^\dagger \left[ V^\dagger H'_{\text{BdG}}(0) V \right] \mathbf{\Gamma} . \tag{29}$$

The matrix elements of $H'_{\text{BdG}}(0)$ are related to the dispersion of the quasiparticle levels with the phase across the junction, and hence to the current carried by these levels. The perturbation $\delta H$ introduces a coupling between the states $|m, \vec{v}\rangle$ and $|m', \vec{v}'\rangle$ when $m' - m = \pm 1$, with a matrix element proportional to $\langle \vec{v} | \mathbf{\Gamma}^\dagger [V^\dagger H'_{\text{BdG}}(0) V] \mathbf{\Gamma} | \vec{v}' \rangle$.

The problem can now be easily tackled numerically, solving the Hamiltonian $H_0 + \delta H$ with a truncated basis consisting of low-energy many-body states. The spectral function $S_N(\omega)$ of Eq. (8) can also be computed numerically using that, in the harmonic limit, $N = i(a^\dagger - a)/\sqrt{z}$. The spectrum and the spectral function calculated for the finite-length junction model (see Sec. 4.3) using this approach are plotted in Fig. 7, for the same model parameters as in Fig. 6.

Comparing Figs. 6 and 7, we find that many of the qualitative features appearing in the spectral function are reproduced within the harmonic expansion. In particular, the characteristic behavior of the spectral function near avoided level crossings can be easily understood.

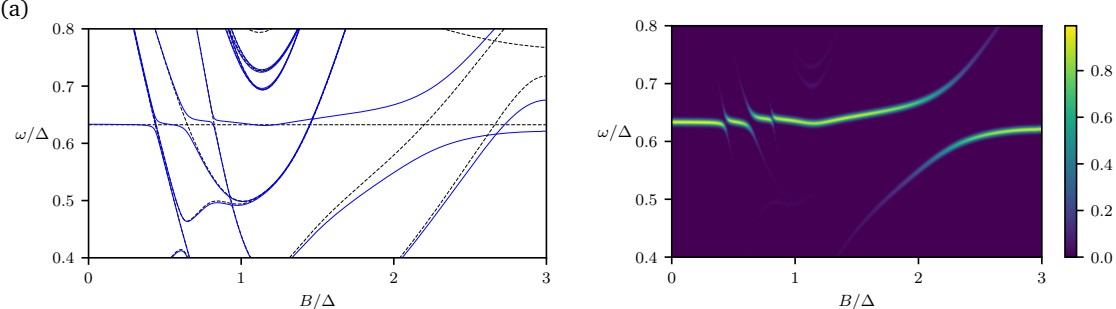

(a)

Figure 7: (a) Excitation spectrum and (b) spectral function obtained using the harmonic expansion for a finite-length junction with the same parameters as in Fig. 6. The black dashed lines in (a) correspond to the excitation energies of the unperturbed Hamiltonian, while the blue solid lines are obtained including the perturbation to lowest order in $\phi$. The truncated basis used here consists of two lowest energy harmonic oscillator states and two-quasiparticle fermionic states constructed from four lowest energy single-particle states. For presentation purposes we plot the spectral function in (b) convolved with a Gaussian with $\sigma = 5 \times 10^{-3}$.

For concreteness, consider a crossing between the unperturbed levels $|1, \vec{0}\rangle$ and $|0, \vec{\nu}\rangle$ as the magnetic field $B$ is tuned through some value $B_0$. Near the crossing, we may approximate the unperturbed energies as $E_{1,\vec{0}}(B) \approx \omega_p^0$ and $E_{0,\vec{\nu}}(B) \approx \omega_p^0 + 2\lambda(B - B_0)$, where $2\lambda \equiv E'_{0,\vec{\nu}}(B_0)$. Including the perturbation $\delta H$ and solving the resulting two-level problem, we obtain hybridized eigenstates of the form $|\pm\rangle = a_\pm(B)|1, \vec{0}\rangle + b_\pm(B)|0, \vec{\nu}\rangle$ with energies

$$E_\pm(B) = \omega_p^0 + \lambda(B - B_0) \pm \sqrt{\lambda^2(B - B_0)^2 + z\zeta^2} \,, \tag{30}$$

where $\zeta = |\langle \vec{0}|\mathbf{\Gamma}^\dagger[V^\dagger H'_{\mathrm{BdG}}(0)V]\mathbf{\Gamma}|\vec{\nu}\rangle|$. The brightness of the spectral line corresponding to the transition $|0\rangle \to |\pm\rangle$, where $|0\rangle$ denotes the ground state, is proportional to $|\langle\pm|N|0\rangle|^2$. This matrix element is just $|a_\pm(B)|^2/z$:

$$|\langle\pm|N|0\rangle|^2 = \frac{1}{2z}\left(1 \mp \frac{\lambda(B - B_0)}{\sqrt{\lambda^2(B - B_0)^2 + z\zeta^2}}\right). \tag{31}$$

The intensity of the two spectral lines is thus identical at the degeneracy point, $B = B_0$, and becomes more and more asymmetrical away from the degeneracy point.

At the same time, since Fig. 6 was obtained for $E_J^0/E_c = 5$, i.e. not very deep in the harmonic limit, some features are not captured correctly. First, in Fig. 7 the plasma frequency is higher than Fig. 6. This discrepancy, which also shifts the exact positions of some of the avoided crossings, can be explained by the anharmonic correction to the plasma frequency that is brought by the fourth-order expansion of $-E_J^0 \cos\phi$, $\delta\omega_p^0 \approx -E_c$ [4], and which is automatically included in the MPS simulations of Sec. 4. Second, we note the level crossing at $B/\Delta \approx 2.7$ and $\omega/\Delta \approx 0.6$ in Fig. 7, which is avoided in Fig. 6 (near $B/\Delta \approx 2.3$ and $\omega/\Delta \approx 0.5$). The fact that this crossing is not avoided within the harmonic expansion can be understood as follows. Since $\phi$ appears in $H_{\mathrm{BdG}}(\phi)$ only in the phases of the terms that hop fermions between the left and right parts of the wire, the perturbative couplings in $\delta H$ only involve fermion quasiparticle levels with support at the junction and are thus local. On the other hand, we find that the two states involved in the crossing differ in the occupation of fermion modes far away from the junction and thus cannot be coupled in the harmonic expansion. In Sec. 5.2, we will discuss how the avoided crossing can be understood in terms of non-perturbative effects. To show that both discrepancies are due to the relatively low

value of $E_J^0/E_c$, in Appendix C we compare the spectral function obtained using the exact MPS simulation and the one obtained using the harmonic expansion for a higher value $E_J^0/E_c = 20$, and show that there is an excellent agreement between the two.

Finally, we note that in principle it should be possible to systematically improve the perturbative expansion in $z$ by including the higher-order terms which were so far neglected in Eq. (26) as well as in the expansion of the cosine potential. In general, the $n$-th order term introduces a coupling between the states $|m, \vec{\nu}\rangle$ and $|m', \vec{\nu}'\rangle$ with $|m' - m| \leq n$. In addition, it can cause a shift in the eigenvalues, even away from level crossings. However, in order to be consistent, the expansion would have to take into account the contribution of the low-energy fermionic degrees of freedom to the plasma frequency as well as the coupling between the low- and high-energy degrees of freedom which were separated in Eq. (7); thus, it appears to be a challenging approach.

## 5.2 Non-perturbative couplings

We now address the non-perturbative couplings that are not captured within the harmonic expansion. As mentioned in passing in Sec. 3, non-perturbative corrections arise due to quantum phase slips between the minima of the potential energy at $\phi = 0$ and $\phi = 2\pi$. In the case of the standard qubit Hamiltonian $E_c(N - N_g)^2 - E_J \cos \phi$, it is well-known that, in the limit $E_J \gg E_c$, quantum phase slips give rise to the charge dispersion of the energy levels with magnitude $\propto e^{-\sqrt{8E_J/E_c}}$ [4]. This effect is, for instance, visible in the energy spectra of Fig. 3(b). These corrections are non-perturbative, as manifested by their singular dependence $\propto e^{-c/z}$ on the expansion parameter $z$ of the previous Section.

In this Section, we examine in detail an example for how these non-perturbative corrections can prominently affect the microwave response in the topological phase. In particular, we show that non-perturbative effects can resolve level crossings that are not resolved perturbatively, leading to the appearance of avoided crossings in the spectral response. The basic physics of this phenomenon is as follows: consider a level crossing between two given states $|e_+\rangle$ and $|f_+\rangle$, as in Fig. 8(a), which are not coupled by perturbative terms local at the junction. If non-perturbative effects cause $|e_+\rangle$ to hybridize with a third state $|e_-\rangle$ that *does* couple perturbatively to $|f_+\rangle$, this will lead to a *non-perturbative avoided crossing*, as seen in Fig. 8(b).

As a concrete example, we extend the minimal model presented in Sec. 3 to include an additional fermionic mode localized (for simplicity) on the right island. We assume that this mode couples to the Majorana mode on the left island via single-particle tunneling across the junction. The Hamiltonian is then given by

$$H = H_J + H_M + H_\epsilon + H_\zeta , \tag{32}$$

where

$$H_J = E_c N^2 - E_J \cos \phi , \tag{33a}$$

$$H_M = i\gamma_2\gamma_3 E_M \cos(\phi/2) , \tag{33b}$$

$$H_\epsilon = \epsilon\, c^\dagger c , \tag{33c}$$

$$H_\zeta = i\zeta\,\gamma_2\big(c\,e^{i\phi/2} + \text{h.c.}\big) , \tag{33d}$$

and we have implicitly set $N_g = 0$. Here, $c^\dagger$ ($c$) is the creation (annihilation) operator for the extra fermionic mode, $\epsilon$ is its energy, and $\zeta$ is its coupling strength to the Majorana mode on the left island. We assume that the energy $\epsilon$ is comparable to the plasma frequency $\omega_p$, and that it can be tuned by an external parameter (such as the Zeeman energy $B$, as in the previous section). Writing the additional fermion operator in terms of Majorana operators,

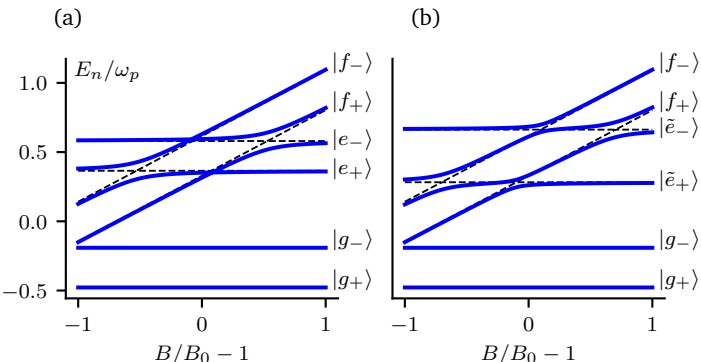

Figure 8: Spectrum of the Hamiltonian (32) projected onto the low-energy subspace, for $E_J/E_c = 5$, $E_M/E_c = 1$. We take $\epsilon(B) = \bar{\omega} + \lambda(B/B_0 - 1)$ with $\lambda/E_c = 3$ and the coupling $\zeta = 0.1$. Black dashed lines correspond to the spectrum of $\tilde{H}$, while the blue solid lines correspond to the spectrum of $\tilde{H} + \tilde{H}_\zeta$. In (a) we artificially set the magnitude of the charge dispersion $\delta_1$ to zero, thus neglecting all non-perturbative corrections, while in (b) the spectrum with finite $\delta_1$ is plotted. Once non-perturbative corrections are taken into account, hybridization between $|e_+\rangle$ and $|e_-\rangle$ (due to finite charge dispersion) results in the avoided crossing between the hybridized states $|\tilde{e}_\pm\rangle$ and $|f_\pm\rangle$.

$c = (\gamma_5 + i\gamma_6)/2$, we rewrite $H_\zeta$ as

$$H_\zeta = i\zeta\,\gamma_2\big(\gamma_5\cos(\phi/2) - \gamma_6\sin(\phi/2)\big)\,. \tag{34}$$

Analogously to the basis defined in Eq. (21) that was used to study the four-Majorana model in Sec. 3, we can define a basis for the full Hilbert space of this model that satisfies the parity constraint (16) and diagonalizes $H_J + H_\epsilon$:

$$|m; n_{12}, n_{34}, n_c\rangle = |m, (-1)^{n_{12}}\rangle_b \otimes |n_{12}, n_{34}, n_c\rangle\,. \tag{35}$$

Here, $|m, \pm\rangle_b$ are the bosonic states defined in Sec. 3 and correspond to the $m$-th energy eigenstate of $H_J$ with even/odd bosonic parity and energy $E_{m,\pm}$; $n_{12}$ and $n_{34}$ are the occupation numbers encoded in the MZMs of the left and right island, $n_{ij} = (1 + i\gamma_i\gamma_j)/2$; and $n_c$ is the occupation number of the extra fermionic mode.

In the harmonic limit $E_J/E_c \to \infty$, $\delta_m \equiv E_{m,+} - E_{m,-} \to 0$ and thus, for each $m$, the states with different $n_{12}$ but equal $n_c$ become degenerate eigenstates of $H_J + H_\epsilon$. Furthermore, in this limit, $H_M$ has non-vanishing matrix elements only between states with the same $m$. Restricting our discussion to low energies (namely, to states with energies up to order $\omega_p$), in the harmonic limit the eigenstates of $H_J + H_\epsilon + H_M$ are given by:

$$|g_\pm\rangle \equiv \frac{|0;000\rangle \pm |0;110\rangle}{\sqrt{2}}\,, \tag{36a}$$

$$|e_\pm\rangle \equiv \frac{|1;000\rangle \pm |1;110\rangle}{\sqrt{2}}\,, \tag{36b}$$

$$|f_\pm\rangle \equiv \frac{|0;011\rangle \pm |0;101\rangle}{\sqrt{2}}\,. \tag{36c}$$

At a large but finite $E_J/E_c$, the energy levels of $H_J$ acquire a finite charge dispersion,

$$|\delta_m| \sim E_c\,(E_J/E_c)^{\frac{m}{2}+\frac{3}{4}}\,e^{-\sqrt{8E_J/E_c}}\,. \tag{37}$$

This non-perturbative energy splitting grows with $m$, allowing us to consider for simplicity the regime $|\delta_0| \ll |\delta_1| \approx E_M$, and to neglect $\delta_0$. Projecting the first three terms of the Hamiltonian (32) onto the basis (36), we obtain (up to a constant shift)

$$\tilde{H} = H_g + H_e + H_f \,, \tag{38}$$

with

$$H_g = -\eta_{0,0} E_M \left( |g_+\rangle\langle g_+| - |g_-\rangle\langle g_-| \right) , \tag{39a}$$

$$\begin{aligned}
H_e = \quad & \bar{\omega} \left( |e_+\rangle\langle e_+| + |e_-\rangle\langle e_-| \right) \\
& -\eta_{1,1} E_M \left( |e_+\rangle\langle e_+| - |e_-\rangle\langle e_-| \right) \\
& +\delta_1 \left( |e_+\rangle\langle e_-| + |e_-\rangle\langle e_+| \right) ,
\end{aligned} \tag{39b}$$

$$\begin{aligned}
H_f = \quad & \epsilon(B) \left( |f_+\rangle\langle f_+| + |f_-\rangle\langle f_-| \right) \\
& -\eta_{0,0} E_M \left( |f_+\rangle\langle f_+| - |f_-\rangle\langle f_-| \right) .
\end{aligned} \tag{39c}$$

Here, $\eta_{m,m}$ are the overlap integrals defined in Eq. (23) and $\bar{\omega}$ is the average energy difference between the $m = 0$ and $m = 1$ bosonic states, $\left[ (E_{1,+} - E_{0,+}) + (E_{1,-} - E_{0,-}) \right]/2$. In the harmonic limit $\bar{\omega} \to \omega_p$. We see that $|g_\pm\rangle$ and $|f_\pm\rangle$ are still eigenstates of $\tilde{H}$, but $H_e$ mixes $|e_\pm\rangle$; its eigenstates, to lowest order in $q \equiv \delta_1/(2\eta_{1,1} E_M)$, are given by

$$|\tilde{e}_+\rangle \approx |e_+\rangle - q|e_-\rangle \,, \tag{40a}$$

$$|\tilde{e}_-\rangle \approx |e_-\rangle + q|e_+\rangle \,. \tag{40b}$$

At this point, we can understand why, as pointed out in Sec. 3, the two spectral lines in the topological phase are only visible away from the harmonic limit. To this end, note that the charge operator $N$ couples $|g_+\rangle$ to $|e_+\rangle$ but not to $|e_-\rangle$; on the other hand, it couples $|g_+\rangle$ to both $|\tilde{e}_+\rangle$ and $|\tilde{e}_-\rangle$.

Finally, we consider a finite coupling $\zeta$. Projecting $H_\zeta$ onto the basis (36), we obtain

$$\tilde{H}_\zeta = i\zeta\,\chi_{0,1}\big( |f_+\rangle\langle e_-| - |f_-\rangle\langle e_+| \big) + \text{h.c.} \,, \tag{41}$$

where, similarly to Eq. (23), $\chi_{m,m'}$ is the overlap integral

$$\chi_{m,m'} = \int_0^{4\pi} d\phi \, \psi_{m,+}^*(\phi) \sin(\phi/2) \, \psi_{m',-}(\phi) \,, \tag{42}$$

with $\psi_{m,\pm}(\phi) \equiv \langle\phi|m,\pm\rangle_b$. Note that $\tilde{H}_\zeta$ only couples $|e_+\rangle$ to $|f_-\rangle$ and $|e_-\rangle$ to $|f_+\rangle$. Thus, in the harmonic limit, or more specifically for vanishing $\delta_1$, there is no avoided crossing between $|e_+\rangle$ and $|f_+\rangle$, as can be seen in Fig. 8(a). Away from the harmonic limit, the magnitude of the avoided crossing between $|\tilde{e}_+\rangle$ and $|f_+\rangle$ is

$$|\langle\tilde{e}_+|\tilde{H}_\zeta|f_+\rangle| \approx \frac{\zeta\,\chi_{0,1}\,\delta_1}{2\eta_{1,1} E_M} \,. \tag{43}$$

This avoided crossing is clearly visible in Fig. 8(b). It is manifestly non-perturbative, and vanishes in the harmonic limit.

# 6 Conclusions and Outlook

We have carefully studied the response of a capacitively shunted topological Josephson junction, using a combination of accurate numerical techniques and theoretical approaches that

allow us to incorporate a microscopic description of the topological phase. Our results indicate that detecting the signatures of the topological phase in the transmon-like setup of Fig. 1, as proposed for instance in Ref. [60] and also as described in Sec. 3, can be complicated by two factors. First, we do not find strong signatures of the topological transition itself on the simulated frequency spectra of the junction. Second, the presence of non-topological Andreev bound states can hinder the signatures of coupled Majorana zero-modes at the Josephson junction in the measurable low-frequency spectrum. Our simulations show that this second problem becomes particularly relevant in parameter regimes that tend to have a higher density of sub-gap states that couple to the phase degree of freedom.

For a quantitative understanding of experiments performed on nanowire Josephson junctions, including the electrostatic environment and the presence of multiple sub-bands can be very important. While the numerical calculations presented here are based on an effective one-band model of a Majorana nanowire, the methods themselves can in principle be applied to more realistic models. This is particularly true for the perturbative approach described in Sec. 5, which can take as input low-energy BdG spectra obtained from large-scale microscopic simulations of realistic devices. For future research, it would also be valuable to find a way to systematically include the contribution of quantum phase slips without resorting to an intensive MPS calculation, and to consider the case in which the nanowire Josephson junction hosts a quantum dot with finite charging energy.

## Acknowledgements

The authors would like to thank Leonid Glazman, Roman Lutchyn, Dmitry Pikulin, Thorvald Larsen, Gijs de Lange and Angela Kou for insightful discussions. CM and BvH thank William Cole and Chetan Nayak for collaboration on related projects. AK's research at KITP is supported by the Gordon and Betty Moore Foundations EPiQS Initiative through Grant GBMF4304.

## A  Tight-binding model

The tight-binding Hamiltonian describing the spin-orbit coupled nanowire forming the Josephson junction used in the simulations is given by

$$H_{\text{BdG}}(\phi) = H_{\text{SC,L}} + H_{\text{J}}(\phi) + H_{\text{SC,R}} \,, \tag{44}$$

where $H_{\text{SC,L(R)}}$ describe the left (right) regions of the wire that are proximity coupled to a superconductor, and $H_{\text{J}}$ describes the junction.

For concreteness, we write down the Hamiltonian after the gauge transformation defined in Eq. (9). The Hamiltonian in the superconducting regions is given by

$$
\begin{aligned}
H_{\text{SC,L(R)}} = &-\sum_{i=i_0,s,s'}^{i_0+N_x-2} \left[ c_{i,s}^\dagger (t\delta_{s,s'} + i\nu\sigma_{s,s'}^y) c_{i+1,s'} + \text{h.c.} \right] \\
&+ \sum_{i=i_0,s,s'}^{i_0+N_x-1} c_{i,s}^\dagger \left( -(\mu-2t)\delta_{s,s'} + B\sigma_{s,s'}^x \right) c_{i,s'} \\
&+ \sum_{i=i_0}^{i_0+N_x-1} \left( \Delta c_{i,\uparrow} c_{i,\downarrow} + \text{h.c.} \right) \,,
\end{aligned}
\tag{45}
$$

where $i_0 = 1$ ($i_0 = N_x + W + 1$) for the Hamiltonian describing the left (right) region, and $W$ denotes the number of the sites in the junction. The hopping amplitude, $t$, and the spin-orbit

coupling, $v$, are related to the continuum parameters by $t = 1/(2ma^2)$ and $v = \alpha/a$, where $a$ is the lattice constant. Here $\sigma^{x,y}$ are the respective Pauli matrices.

In the main text, we consider two different models for the junction. For the short junction (weak link) model studied in 4.2, $W = 0$ and the Hamiltonian for the junction is given by,

$$H_J = -\kappa e^{i\phi/2} \sum_{s,s'} c^\dagger_{N_x,s}(t\delta_{s,s'} + iv\sigma^y_{s,s'})c_{N_x+1,s'} + \text{h.c.} , \tag{46}$$

where $N_x$ is the rightmost site of the left superconducting region, and $N_x + 1$ is the leftmost site of the right superconducting region.

For the finite-length junction model studied in 4.3, the Hamiltonian for the junction is

$$\begin{aligned}
H_J = &-\kappa_L \sum_{s,s'} \Big[ e^{i\phi/2} c^\dagger_{N_x,s}(t\delta_{s,s'} + iv\sigma^y_{s,s'})c_{N_x+1,s'} + \text{h.c.} \Big] \\
&- \sum_{i=N_x+1,s,s'}^{N_x+W-1} \Big[ c^\dagger_{i,s}(t\delta_{s,s'} + iv\sigma^y_{s,s'})c_{i+1,s'} + \text{h.c.} \Big] \\
&+ \sum_{i=N_x+1,s,s'}^{N_x+W} c^\dagger_{i,s}(-(\mu-2t)\delta_{s,s'} + B\sigma^x_{s,s'})c_{i,s'} \\
&- \kappa_R \sum_{s,s'} \Big[ c^\dagger_{N_x+W,s}(t\delta_{s,s'} + iv\sigma^y_{s,s'})c_{N_x+W+1,s'} + \text{h.c.} \Big] .
\end{aligned} \tag{47}$$

## B  Extracting $E_M$

In order to extract an effective $E_M$ from the tight binding model of the junction, we diagonalize the BdG Hamiltonian and calculate the energies of the ground state and first excited state as functions of the phase $\phi$ defined on a $4\pi$-periodic domain (see Fig. 9). In a Josephson junction geometry and for an infinite wire, the model of Eq. (24) would give an Andreev level crossing of at $\phi = \pi + 2\pi n$ (where $n \in \mathbb{Z}$), leading to the $4\pi$-periodicity of the ground state. For a finite system, the crossings will be avoided, with the splitting determined by the coupling between the Majorana modes at the junction and those at the far ends of the wire. Nevertheless, If the system is deep in the topological phase (in practice we consider a Zeeman energy of magnitude $B = 2B_c$), the avoided crossings at $\phi = \pi, 3\pi$ will be small enough to allow us to define a $4\pi$-periodic ground state energy, $E_{4\pi}(\phi)$. An effective $E_M$ is then defined as

$$E_M \equiv \frac{1}{2\pi} \int_0^{4\pi} E_{4\pi}(\phi)\cos(\phi/2) . \tag{48}$$

## C  Numerical results in the harmonic limit

In Fig. 10 we plot the spectral function obtained from the MPS simulations in the harmonic limit, $E_J^0/E_c = 20$, and the spectral function obtained using the harmonic expansion for the same parameters. To highlight the agreement between the two for the strength of the couplings between the plasma mode and the fermionic quasiparticle excitations, we take into account the anharmonic corrections that shift the energy of the first excited state with respect to the plasma frequency in the harmonic limit $\omega_p = \sqrt{8E_J E_c}$, by $\delta\omega_p \approx -E_c$. This is achieved by including the next order term in the expansion of the cosine potential, namely $-E_J\phi^4/24 = -E_J z^2(a + a^\dagger)/24$, as part of the bosonic Hamiltonian $H_{0,b}$, given in Eq. (25).

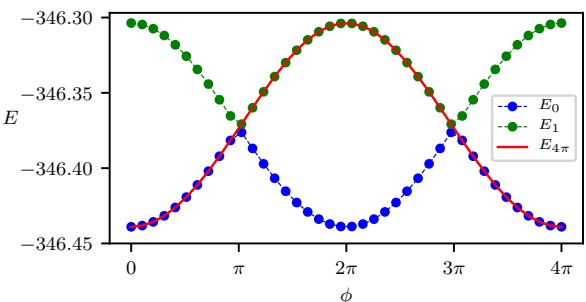

Figure 9: Energies of the ground state ($E_0$; blue dashed line), and the first excited state ($E_1$; green dashed line), obtained using the BdG spectrum, calculated for the weak-link tight binding model (see Appendix A). Tight binding parameters used are the same as in Fig. 4, with a Zeeman energy $B = 2B_c = 2$. The approximate crossings in the spectrum at $\phi = \pi, 3\pi$ allow us to define a $4\pi$-periodic ground state energy $E_{4\pi}$, plotted as the solid red line.

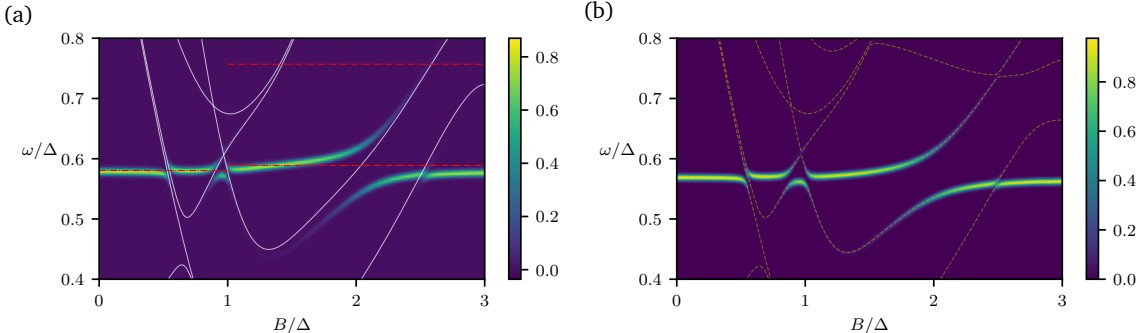

Figure 10: Spectral functions obtained using (a) MPS simulations and (b) the harmonic expansion (taking into account the anharmonic corrections to the plasma frequency), for a finite-length junction model, as a function of the Zeeman energy $B$ for $N_g = 0$. Junction parameters in units of $\Delta$ are $E_c = 0.05$, $E_J^0 = 1$, and wire tight-binding parameters are $t = 1.5, v = 2, \mu = 0, \kappa_L = \kappa_R = 0.3$. The length of each part of the wire that is coupled to a superconductor is $N_x = 20$, and the length of the junction is $W = 6$. (a) Red dashed lines, plotted on top of the spectral function for $B < B_c$ ($B > B_c$), are the spectral lines expected deep in the trivial (topological) phase. White solid lines are the energies of 2-quasiparticle excitations obtained from the BdG Hamiltonian for a phase difference of $\phi = 0$ between the superconducting islands. (b) Dashed lines correspond to the excitation spectrum obtained within the perturbative approach. For presentation purposes a convolution with a Gaussian with $\sigma = 5 \cdot 10^{-3}$ is performed.

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
