# Peer review of "Spectral response of Josephson junctions with low-energy quasiparticles"

_SciPost Physics, doi:SciPost Phys. 7, 050 (2019)_

## Round 1 · Referee Report · Anonymous · 2019-8-19

Strengths

(1) Timely. The manuscript focuses on a timely and technologically important problem
(2) The authors' numerical approach allows for an easy extension to realistic models which would help the ongoing experimental effort.
(3) Simple analytical models that help clarify the physics coming out of the numerical simulations.
(4) Clear introduction and exposition; I enjoyed reading it.

Weaknesses

(1) In my opinion, the weakest point, if it can be seen as one, is that the manuscript's lack of a strong punchline. However, its main result that the junctions are too complicated to distinguish topological transitions in a transmon setup is definitely worthy of publication.

Report

The present manuscript focuses on an interesting problem: the dynamics of a Josephson junction in the presence of low lying quasiparticles (particularly that formed by a pair of Majorana bound states). The authors focus on a transmon setup and calculate both numerically and analytically the spectral function of the charge operator as well as the excitation spectra of the junction across the topological transition. Presence of Andreev bound states of nontopological origin, a likely complication, is also considered. The numerical method adopted by the authors seems extendable to much more realistic models which would increase the relevance of the present work to ongoing experimental effort. As I mention above, the main result that the Josephson junctions might be too complicated to observe distinguishing signatures of topological transitions in a transmon setup is definitely worth publishing. Therefore I recommend publication after the minor revisions decribed below are done.

Requested changes

Fig 3c: supply legend for the colored-dashed lines

I would like to ask the authors to consider publishing their code.

---

## Editorial Decision

published